# Can individual subjective confidence in training questions predict group performance in test questions?

**Masaru Shirasuna** *, **Hidehito Honda**

Faculty of Psychology, Otemon Gakuin University, Ibaraki-shi, Osaka, Japan

* m.shirasuna1392@gmail.com

**Data Availability Statement:** All data, code, and materials used in our experiments are available at OSF: https://osf.io/rj4dv/.

**Funding:** The present study was funded by Japan society for the promotion of science (JSPS)

## Abstract

When people have to solve many tasks, they can aggregate diverse individuals' judgments using the majority rule, which often improves the accuracy of judgments (wisdom of crowds). When aggregating judgments, individuals' subjective confidence is a useful cue for deciding which judgments to accept. However, can confidence in one task set predict performance not only in the same task set, but also in another? We examined this issue through computer simulations using behavioral data obtained from binary-choice experimental tasks. In our simulations, we developed a "training-test" approach: We split the questions used in the behavioral experiments into "training questions" (as questions to identify individuals' confidence levels) and "test questions" (as questions to be solved), similar to the cross-validation method in machine learning. We found that (i) through analyses of behavioral data, confidence in a certain question could predict accuracy in the same question, but not always well in another question. (ii) Through a computer simulation for the accordance of two individuals' judgments, individuals with high confidence in one training question tended to make less diverse judgments in other test questions. (iii) Through a computer simulation of group judgments, the groups constructed from individuals with high confidence in the training question(s) generally performed well; however, their performance sometimes largely decreased in the test questions especially when only one training question was available. These results suggest that when situations are highly uncertain, an effective strategy is to aggregate various individuals regardless of confidence levels in the training questions to avoid decreasing the group accuracy in test questions. We believe that our simulations, which follow a "training-test" approach, provide practical implications in terms of retaining groups' ability to solve many tasks.

## 1. Introduction

### 1.1 Dealing with uncertainty by aggregating diverse judgments: Wisdom of crowds

When people try to solve tasks such as binary-choice inferential questions, they often experience "uncertainty" because of their lack of knowledge and do not know which alternative to

KAKENHI No. 18H03501 and No. 22H03915 (to HH). The funder had no role in study design, data collection and analysis, decision to publish, or preparation of the manuscript.

**Competing interests:** The authors declare no competing interests.

choose. To reduce such uncertainty and boost accuracy, people can take advantage of more than one individual's opinions and make judgments based on the majority rule. Previous studies showed that simply aggregating two or more people's judgments by, for example, using a majority rule or averaging numerical estimations (i.e., group judgments) can often be more accurate than one person's judgment. This effect, known as the *wisdom of crowds* effect [1–8], has been observed in various fields including general knowledge tasks [9, 10], forecasts [11, 12], and matter of taste [13].

Especially in inferences of binary choice situations, when the mean of individual accuracy is more than 0.5 and individuals make "diverse" judgments (i.e., different individuals make different mistakes), the group is likely to have the benefits of a majority rule (for hypothetical examples, see Fig 1; see also [1]). For example, in binary choice tasks, if individuals make different (diverse) judgments individually, errors are likely to cancel each other out at the collective level [1, 14]. In contrast, if individuals make exactly the same non-diverse judgments, the majority rule will not improve the accuracy of the group judgment because errors will remain at the collective level [1].

## 1.2. Subjective confidence in "training" questions as a cue for predicting accuracy in "test" questions

Considering which opinion to accept in uncertain situations where the correct answer is unclear, individuals' subjective confidence regarding the current question(s) (i.e., how confident they feel about answering correctly) can be a useful cue for identifying who has or has not high abilities for the task (see also [15, 16]). Because subjective confidence can generally well predict their task accuracy [17–22], it seems that group judgments will perform well if individuals with high confidence are aggregated.

However, in the real world, one must often deal with not only one task set, but also another different task set. In this study, we refer to these two task sets as "training questions" and "test questions," respectively, in accordance with the cross-validation approach (described later). To

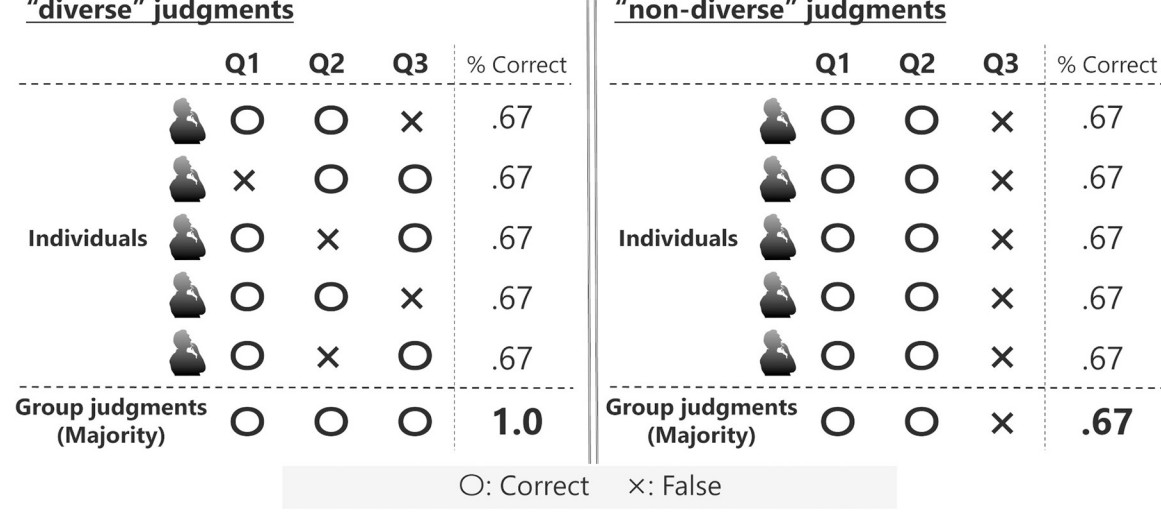

**Fig 1. Hypothetical examples of the wisdom of crowds by a majority rule.** These examples assume that five individuals respond to three binary choice questions, and they make a group judgment using a majority rule for each question. Left: Individuals make different judgments individually (i.e., "diverse" judgments), which will cancel each other out at the collective level. Right: Individuals make the same judgments individually (i.e., "non-diverse" judgments), which will not improve the accuracy of judgments even at the collective level. These examples suggest the importance of the diversity of judgments in a group to achieve the wisdom of crowds.

what extent can individuals' confidence in training question(s) predict groups' performance in other, test questions? Even if a person successfully solved one training question with high confidence, it is unclear whether he/she can also solve other test questions. For example, even if Person X could make correct judgments with high confidence and Person Y could not in Question *i*, Person Y may make correct judgments with high confidence and Person X may not in Question *j*. Although many findings about individuals' subjective confidence in their judgments have been reported, previous studies focused on individuals' past judgments: People considered how (in)accurate a judgment they had already made would be in a certain question. Therefore, it is unclear whether individuals' confidence in a certain training set can predict their performance in another as yet unseen test set. Possibly, even if individuals solved one or a few training question(s) with high confidence, this success might merely be a coincidence and they do not have enough knowledge to solve further questions. Furthermore, many studies have also demonstrated that people sometimes overestimate their chance of success or actual abilities regarding the domain of the current tasks (i.e., overconfidence [23–31]). Based on these considerations, even if the accuracy of group judgments made by individuals with high confidence in one or a few training question(s) is very high, this may decrease for the test questions (see also [32]).

A recent study [33] focused on similar situations, investigating whether and how knowledge of the old environment could be generalized to new environments in terms of an information search through social interactions. This study demonstrated that, through computational analyses of participants' behaviors in a multi-armed bandit task, paired participants performed a more efficient information search than solo participants. This improvement was significantly correlated with their subjective understanding of the generative rule (i.e., estimations of hyperparameters that created search spaces in the experiments). Although this study provided important insights and viewpoints, it did not focus on subjective confidence. Thus, it remains unclear whether and to what extent individuals' training confidence can become a useful predictor of the test performance regarding group judgments.

## 1.3 Mixing various confidence levels in one group: A possible strategy to avoid decreasing accuracy in test questions

To deal with uncertain situations (i.e., solving both training and test questions) by the majority rule, we propose a strategy of aggregating various individuals in terms of confidence levels. Even if a group successfully solves the training questions, it is unclear whether this group will also be able to solve the test questions. Possibly, individuals with extremely high confidence in some question(s) could merely "by chance" solve these question(s) and are in fact, unfamiliar with further questions. In other words, the regression to the mean may occur from training performance to test performance, or individuals with high confidence may make false judgments or overestimate their own abilities, which is known as overconfidence [23–31]. Therefore, when a group comprises only individuals with high confidence (called the "high-confidence group"), its training accuracy is unlikely to predict its test accuracy. Even if the accuracy of the high-confidence group is very high in the training question(s), it may decrease in the test questions. On the other hand, if individuals with medium level of confidence are added to a group instead of those with high confidence, the abovementioned "by chance" cases are less likely to occur. Therefore, a group comprising both individuals with high and medium confidence for the training questions (called the "mixed-confidence group") may be able to better predict test performance based on their training performance. That is, the mixed-confidence group will show a smaller performance gap between test and training accuracy than the high-confidence groups.

Briefly, we expected that mixing individuals with various subjective confidence levels for the training questions would be an effective strategy to avoid a decrease in the accuracy of the test questions.

## 1.4 Computer simulations using a "training-test" approach

In this study, we theoretically investigated the abovementioned issues to obtain the first evidence of whether and to what extent individuals' confidence levels in training questions can predict their accuracy in test questions. To this end, we performed computer simulations using behavioral data. Specifically, inspired by a cross-validation in machine learning, we developed an approach in which we simulated group judgments in what we called a "training-test" approach: In the computer simulations, we regarded part of (one or a few) the question(s) used in behavioral experiments as training questions and part of the other questions as test questions. Using these two-split task sets, we assumed that individuals were ranked based on their confidence ratings in the training questions. This procedure reflected situations of identifying who had high confidence based on training questions in the real world. When making group judgments, hypothetical groups were generated by collecting individuals based on the ranking of their confidence ratings. In so doing, we manipulated members' confidence levels in a group. We then assumed that these hypothetical groups solved both the training and test questions using a majority rule, and calculated the accuracy of the group judgments. This procedure reflected situations in which individuals made group judgments to deal with both training and test questions in the real world (for more detail on the computer simulations, see section 2.6).

## 1.5 Outline of this study

Modern daily human life (e.g., inferences, consumer choices, portfolio decisions) is characterized by many uncertain situations in which two or more individuals' opinions are needed, and subjective confidence is often used as a cue for considering whose opinion they should accept. Thus, we need to investigate the relationships between individuals' subjective confidence in training questions and the performance of group judgments in test questions.

Because this is the first study to investigate the relationships between individuals' confidence in training questions and group performance in test questions, we aimed to obtain theoretical evidence using simulation-based approaches. The remainder of this paper is organized as follows. First, we conducted behavioral experiments using two types of binary choice tasks: a population inference task and relationships comparison task. In the experiments, we obtained participants' responses and confidence ratings of their own judgments. Using behavioral data, we conducted computer simulations of the group judgments. We assumed that individuals were aggregated based on their confidence in the training questions, and then the group made judgments in the training and test questions using a majority rule.

First, we analyzed behavioral data to obtain insights into group judgments for the training and test questions, and then conducted computer simulations of these group judgments (for details, see Materials and Methods). Specifically,

- Analyses of behavioral data: We first confirmed the relationships between individuals' subjective confidence and their accuracy. We investigated the extent to which subjective confidence could predict accuracy in the same or another question (called "within-" or "between-" questions, respectively).

- Computer simulation 1: We confirmed whether the diversity of individuals' judgmental patterns would differ depending on subjective confidence. Specifically, we ranked individuals

based on confidence ratings in 1 training question and collected 15 individuals from the top ranks (highest), above-median ranks (higher), or all ranks (mixed). We then investigated whether and to what extent judgments accorded between two individuals ($15 \times 14 \times 0.5 = 105$ pairs in total) in nine test questions.

- Computer simulation 2: We investigated the extent to which hypothetical group accuracy in training questions would differ from that in the test questions, manipulating the levels of group confidence (highest, higher, or mixed groups), group size (3 or 15), and number of training questions (1, 5, or 10).

## 2. Materials and methods

In our behavioral experiments, we used two types of tasks, *population inference task* and *relationships comparison task*, to verify the robustness of the experimental results. The experimental protocols in this study conformed to the Declaration of Helsinki and were approved by the Ethics Review Committee for Experimental Research at Otemon Gakuin University.

### 2.1 Participants

Of the 303 participants, 152 were assigned to a population inference task ($n_{male} = 88$, $n_{female} = 62$, $n_{other} = 2$; $M_{age} = 40.5$, $SD_{age} = 9.89$), and 151 to a relationships comparison task ($n_{male} = 86$, $n_{female} = 65$, $n_{other} = 0$; $M_{age} = 43.0$, $SD_{age} = 10.3$). We set the sample sizes as follows: Because this study was the first to investigate the relationships between individuals' confidence in one task set and accuracy in another task set for group judgments, we anticipated medium effect sizes in a one-way analysis of variance (As described later, we focused on the differences between the performance of three hypothetical groups: "highest," "higher," and "mixed"). If the effect size was 0.25, the required sample size was 159; if it was 0.30, the required sample size was 111. Therefore, the sample size was set to approximately 150 for each task. The sample size was calculated using GPower 3.1 [34].

Participants were recruited via a Japanese crowdsourcing service, *Lancers* (https://www.lancers.jp/) in May 2021. All participants provided informed consent before participating in this study. No data that could identify participants were collected. Participants were rewarded with 500 JPY (approximately 4.59 USD in May 2021) for their responses upon completion of the experiment.

### 2.2 Tasks and materials

**Population inference task (Fig 2 upper).**   This task has been used in many studies to examine human inferences, especially in the literature on heuristics [35–39]. In this task, the task structure was that two objects (typically, city names) were presented as alternatives, such as "Which city has a larger population, City A or City B?" Participants were asked to compare these two alternatives and make inferences in terms of population size. We used 70 pairs of city names from a previous study as the experimental stimuli [37]. For the process of selecting the questions and question list, see Supporting Information 1 in S1 File.

**Relationships comparison task (Fig 2 lower).**   This task has recently been proposed to examine human inferences in the context of heuristics (specifically, the adaptive use of heuristics) [40–42]. In this task, the task structure was that objects were presented not only in alternatives but also in a question sentence format. In this task, participants were directly asked general knowledge such as "Which country is City Q in, Country A or Country B?" In this study, we used a relationships comparison task as another task to confirm the consistency and

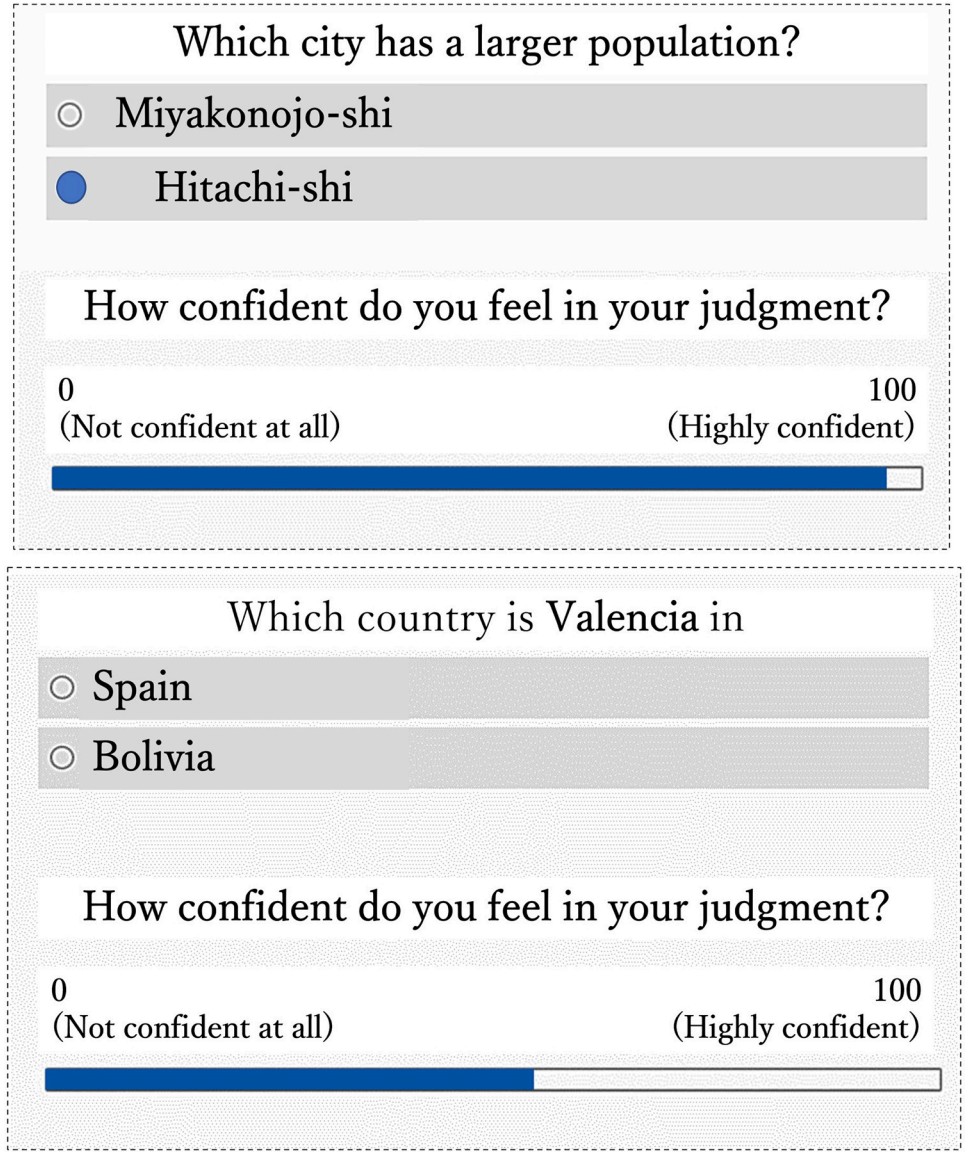

**Fig 2.** Examples of a population inference task (upper) and relationships comparison task (lower). In both tasks, participants were asked to choose one of the two alternatives, and then to rate their confidence for the judgment. Note: In our experiment, all descriptions were displayed in Japanese.

validity of our experimental results. We selected 25 city names based on the rates of correct judgments in the original study as the experimental stimuli [41]. For the process of selecting the questions and the question list, see Supporting Information 2 in S1 File.

Note that, we also conducted other tasks such as those to measure familiarity of each city/country name. For example, for each object, participants rated how familiar they were by using a visual analog scale (0 = not familiar at all; 100 = very familiar). However, the data provided from these additional tasks were not used in the present study. There were more questions for measuring familiarity in a relationships comparison task than in a population inference task. To reduce participants' workload, the numbers of questions between a population inference task and a relationships comparison task differed.

## 2.3 Procedures

All tasks were conducted online, and a graphical user interface (GUI) was created using Qualtrics (https://www.qualtrics.com/jp/). The general procedures were identical for both tasks.

First, participants were presented a question sentence and two alternatives on a computer screen and were asked to make a binary choice (e.g., "Which city has a larger population, Yokohama-shi or Chiba-shi?" in a population inference task; or "Which country is Addis Ababa in, Ethiopia or Egypt?" in a relationships comparison task). Next, they rated their subjective confidence in their judgment (i.e., "How confident do you feel in your judgment?") using a visual analog scale, in which the left and right ends indicate 0 ("not confident at all") and 100 ("highly confident"), respectively. After the binary choice and difficulty rating, the next question was presented on the computer screen. Participants could not skip any questions (i.e., two alternatives forced choice). The order in which the questions and the two alternatives were presented was randomized. Participants assigned to a population inference task answered 70 questions, and those assigned to a relationships comparison task answered 25 questions.

To check whether participants faithfully answered the tasks online, we inserted several questions, called "check problems" (e.g., "Choose 'Japan' in this question. Japan or Tokyo"). As described later, we excluded from the analyses participants who did not correctly respond to the check problems (e.g., chose "Tokyo" in the aforementioned example). The check problems were inserted in the middle of the main task: after the 35th question in a population inference task, and after the 13th question in a relationships comparison task.

## 2.4 Data exclusion

Some participants were excluded from the analyses based on the following criteria. First, we excluded participants who took less than 5 minutes or more than 50 minutes to complete our tasks. Second, we excluded participants who did not answer the check problems as instructed. Consequently, we used data from 150 participants in a population inference task ($n_{\text{male}}$ = 86, $n_{\text{female}}$ = 62, $n_{\text{other}}$ = 2; $M_{\text{age}}$ = 40.6, $SD_{\text{age}}$ = 9.92), and 149 participants in a relationships comparison task ($n_{\text{male}}$ = 85, $n_{\text{female}}$ = 64, $n_{\text{other}}$ = 0; $M_{\text{age}}$ = 43.1, $SD_{\text{age}}$ = 10.3).

## 2.5 Analyses of individuals' behavioral data

We analyzed behavioral data focusing on predictions based on confidence to accuracy "within-"or "between-" questions in individual judgments (Fig 3). In the within-question cases, we predicted individuals' accuracy (correct or false) in one question based on their confidence in the same question. That is, we investigated the extent to which individuals' confidence in Question $i$ could predict their accuracy in Question $i$. We conducted this prediction for all questions (i.e., $i$ = 1, 2, . . ., 70 in a population inference task; $i$ = 1, 2, . . ., 25 in a relationships comparison task). In other words, predictions were made for 70 questions in a population inference task, and 25 in a relationships comparison task.

In between-questions cases, on the other hand, we predicted individuals' accuracy in one question based on their confidence in another question. That is, we investigated the extent to which confidence in Question $i$ could predict accuracy in Question $j$. It may seem unintuitive to focus on between-questions relationships. However, as described earlier, it is unclear whether and to what extent individuals with high confidence for the current tasks (i.e., training questions) can solve additional tasks they face in the future (i.e., test questions). We considered it important to demonstrate this issue both quantitatively and theoretically. Thus, we predicted the accuracy of Question $j$ (as one test question) from the confidence of Question $i$ (as one training question). We conducted this prediction for all question pairs (i.e., $i$ = 1, 2, . . ., 70; $j$ = 1, 2, . . ., 70; $i \neq j$ in population inference task; $i$ = 1, 2, . . ., 25; $j$ = 1, 2, . . ., 25; $i \neq j$ in

**Fig 3. Schematics of the procedure of analyses of behavioral data.** From confidence in Question $i$, we predicted accuracy (correct or false) in Question $i$ in "within-questions" cases or accuracy in Question $j$ ($i{\neq}j$) in "between-questions" cases.

relationships comparison task). In other words, predictions were made for 4,830 (= 70×69) pairs in a population inference task, and 600 (= 25×24) pairs in a relationships comparison task. As described later, one question and three pairs were excluded from analyses in a population inference task because the Rhat of the coefficient of confidence estimated by the Markov Chain Monte Carlo (MCMC) method was above 1.1. Finally, we used 4,689 pairs for analyses in a population inference task.

## 2.6 Computer simulations

As the general procedure for computer simulations, we split the binary choice questions into two task sets: "training questions" and "test questions," inspired by the cross-validation approach. Two computer simulations were performed in this study.

The first simulation evaluated the diversity of individuals' judgmental patterns (Fig 4 upper panel). It is expected that patterns of judgments may differ between individuals depending on their levels of confidence. In a certain question, many individuals with high confidence may tend to make similar judgments, whereas those with low confidence may not show such tendencies. This is because individuals with low confidence will generally have less knowledge than those with high confidence and thus will more frequently make random guesses. If so, judgments observed in the mixed-confidence group are more likely to become diverse than those observed in the high-confidence group. To quantitatively evaluate the diversity of judgments, we calculated the accordance rates between two individuals' judgments in the test questions (i.e., to what extent one individual's judgments accorded with another individual's judgments) and compared the mean of the accordance rates between three confidence categories. Specifically:

i. We randomly selected 10 questions, regarding 1 question as a training question and 9 as test questions.

ii. We ranked (sorted) participants based on their confidence ratings in 1 training question, and then collected 15 participants from one of the three categories: from the 1st to the 15th ("highest"; i.e., 1st, 2nd, . . ., 15th), from the 1st to the median ("higher"; i.e., 1st, 6th, 11th, . . ., 71st), or from the 1st to the lowest ("mixed"; i.e., 1st, 11th, 21st, . . ., 141st).

iii. For each category, we examined the extent of two individuals' judgments in 9 test questions (e.g., if the responses by Participant X and Participant Y were "1 0 0 0 1 1 1 0 1" and

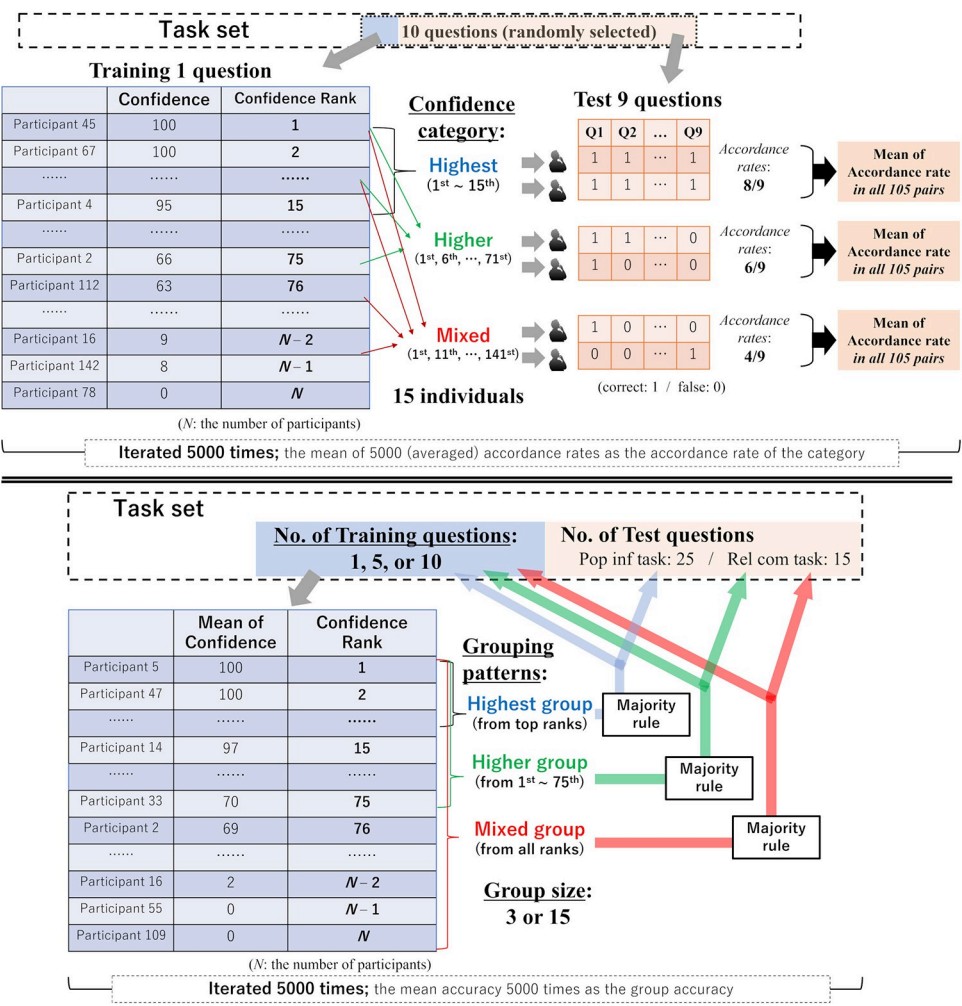

**Fig 4. Schematics of the two computer simulations.** We split the binary choice questions used in behavioral experiments into two sets ("training questions" and "test questions") and iterated the procedures 5,000 times in both simulations. Upper: accordance rates between two individuals' judgments. We collected 15 individuals in 3 patterns (highest, higher, or mixed) based on the confidence rating rankings in 1 training question. For each category, we examined the extent to which one participant's responses accorded with another participant's responses in nine test questions. Lower: group judgments for the training and test questions. We generated hypothetical groups based on individuals' confidence ratings in the training questions. Then, in both the training and test questions, we calculated the accuracy of group judgments by a majority rule. The underlined descriptions denote the conditions we manipulated ("No. of Training questions": 1, 5, or 10. "Group size": 3 or 15. "Grouping patterns": highest group, higher group, or mixed group). For details on the procedure, see the main text.

"1 1 0 1 1 0 1 0 0," respectively, then the accordance rate was 5/9). We averaged the accordance rates for all 105 (= 15×14×0.5) pairs.

iv. We iterated the above steps 5,000 times, and regarded the mean of the 5,000 (averaged) accordance rates as the accordance rate of the category.

The second simulation investigated group performance in the test questions depending on individuals' confidence ratings in the training question(s) (Fig 4 lower panel). We generated hypothetical groups based on individuals' confidence in the training questions, and then examined the extent to which group accuracy would differ between the training and test questions. Specifically:

i. We randomly selected 1, 5, or 10 questions and regarded them as training question(s) (i.e., No. of training questions: 1, 5, or 10). From the other questions, we also randomly selected 25 (in a population inference task) or 15 (in a relationships comparison task) questions and regarded them as test questions.

ii. We calculated the mean of confidence in the training question(s) for each individual and then generated hypothetical groups by selecting 3 or 15 individuals (i.e., group size: 3 or 15). When generating groups, we collected individuals in three categories: from the highest ranks ("highest group"; i.e., collected 1st, 2nd, 3rd individuals, or 1st, 2nd, . . ., 15th individuals), from above-median ranks ("higher group"; i.e., randomly collected 3 or 15 individuals from 1st ~ 75th), or from all ranks ("mixed group"; i.e., randomly collected 3 or 15 individuals regardless of confidence).

iii. In both the training and test questions, we assumed that the group made judgments by the majority rule, and then calculated the rates of correct judgments.

iv. We iterated the above steps 5,000 times, and regarded the mean accuracy 5,000 times as the group accuracy.

## 3. Results

As described earlier, we investigated whether and to what extent individuals' confidence in training questions could predict group performance in test questions through behavioral data analyses and computer simulations. All analyses reported in this paper were conducted using R software (R Core Development Team; version 4.0.5). For the general tendencies of the behavioral data (i.e., each participant's correct/false responses and subjective confidence for each question), see Supporting Information 3 in S1 File. All data, code, and materials used in this study are available at https://osf.io/rj4dv/

### 3.1 Analyses of behavioral data: Predictions of accuracy "within" or "between" questions

We predicted accuracy based on individuals' subjective confidence ratings in the "within-" or "between-" questions (for the detailed procedure, see section 2.5). In the behavioral data analyses, we used R package "brms" [43, 44], applying an MCMC with 3,000 iterations, 1,500 warmups, and 3 chains. We applied a generalized linear model with confidence ratings as an independent variable and correct/false (1/0 dummy) as a dependent variable and then estimated the coefficients of confidence. (For the posterior distributions and trace plots of the coefficients of confidence for each question, see Supporting Information 4 in S1 File). Note that in a population inference task, some questions were excluded from the analyses because the estimations did not successfully converge (Question 37) or because the Rhat of the coefficient of confidence was more than 1.1 in the MCMC ("independent and dependent Question IDs" were "29 and 16," "33 and 3," and "45 and 5," respectively).

We show the results of the predictions in Fig 5, where each point denotes the distribution of the medians of the estimated coefficients of confidence for each question (within-questions; bule) or each pair (between-questions; orange). In both tasks, the mean of the median coefficients tended to be larger in within-questions cases than between-questions cases (Population inference task: mean = 0.016 and SD = 0.019 in within-questions, mean = 0.000 and SD = 0.032 in between-questions. Relationships comparison task: mean = 0.022 and SD = 0.016 in within-questions, mean = 0.008 and SD = 0.010 in between-questions). Furthermore, cases where the Bayesian 95% credible intervals (CI) of the estimated coefficients did

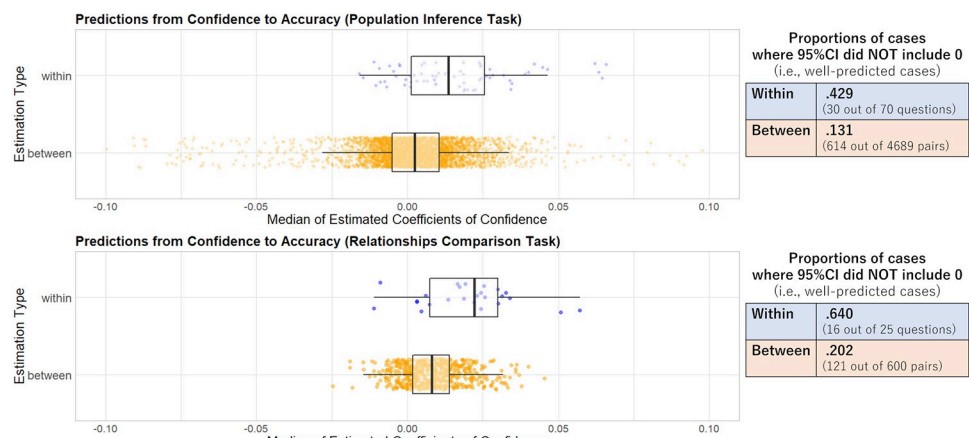

**Fig 5. Distributions of medians of the estimated coefficients of confidence in "within-questions" cases (blue) and "between-questions" cases (orange), and proportions of cases where 95% credible intervals did not include zero (right tables).** Using a generalized linear model, we predicted the accuracy of one question based on participants' confidence ratings in the same question (within-questions), or from those in another question (between-questions). We assumed correct/false (1/0 dummy) as a dependent variable and confidence ratings as the independent variable, and estimated the parameter of the coefficient of confidence using an MCMC method (3,000 iterations, 1,500 warm-ups, and 3 chains). Upper: population inference task. Lower: relationships comparison task.

not include 0 (i.e., confidence could predict accuracy well) were more frequently observed in within-questions than between-questions (Proportions in population inference task: .429 in within-questions and .131 in between-questions. Proportions in relationships comparison task: .640 in within-questions and .202 in between-questions. See the tables on the right in Fig 5). In sum, as predicted, subjective confidence in one question is likely to well predict accuracy in the same question, but unlikely to do so in another question.

### 3.2 Computer simulation 1: Accordance rates of two individuals' judgments in the test questions

In the first computer simulation, we quantitatively evaluated the diversity of individuals' judgment patterns based on their confidence. We ranked individuals by confidence ratings in 1 training question and then calculated the accordance rates between 2 participants' judgments in 9 test questions. This procedure was iterated 5,000 times. (For the detailed procedure, see section 2.6)

In Fig 6, each point denotes the mean accordance rates (average of accordance rates in 105 pairs) for 5,000 iterations, and error bars denote 95% confidence intervals. In both tasks, the order of the mean accordance rates was "highest > higher > mixed" (Population inference task: highest .607; higher .601; mixed .596. Relationships comparison task: highest .573; higher .535; mixed .525). Furthermore, the 95% confidence intervals did not overlap (Population inference task: highest [.605, .609]; higher [.600, .603]; mixed [.594, .597]. Relationships comparison task: highest [.571, .574]; higher [.534, .536]; mixed [.524, .526]). These results indicate that individuals with high confidence tend to make similar and less diverse judgments than those with low confidence. This implies that mixing many individuals in a group regardless of their subjective confidence may lead to diverse judgments and achieve the wisdom of crowds.

### 3.3 Computer simulation 2: Difference in accuracy between training and test questions in group judgments

In the second computer simulation, we simulated the performance of the group judgments. We assumed that hypothetical groups were formed based on individuals' confidence levels in

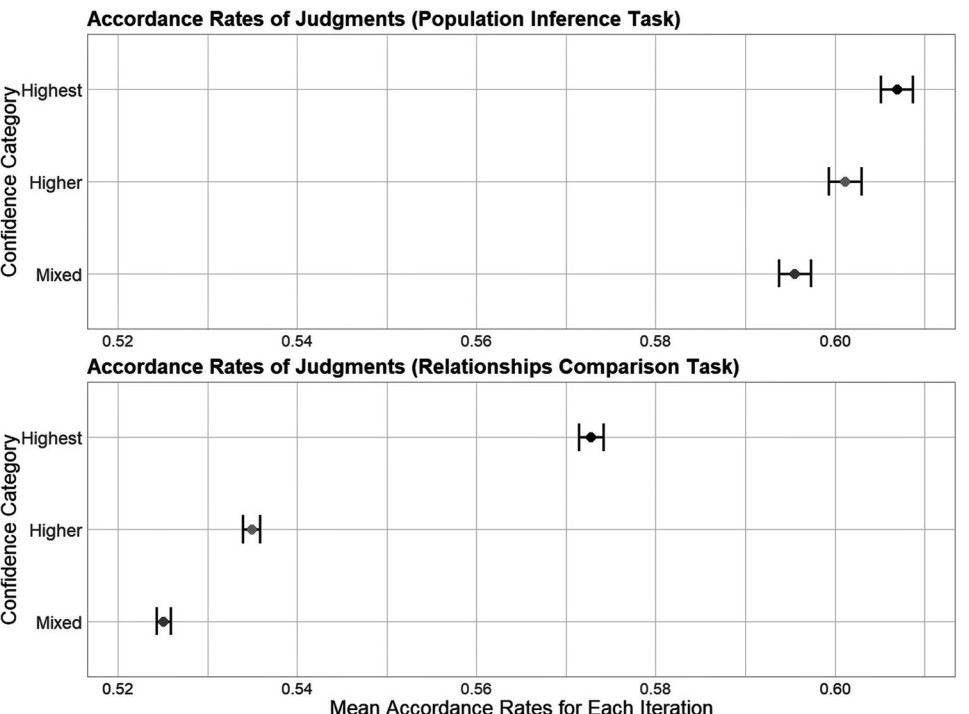

**Fig 6. Accordance rates of two individuals' judgments.** We calculated the mean accordance rates (averaged in 105 pairs) in 5,000 iterations of computer simulations. Each point and the error bars denote the mean of accordance rate and 95% confidence intervals, respectively.

the training questions, and then the groups solved training and test questions using the majority rule. (For details on the procedure and conditions of simulations, see section 2.6). In Fig 7, the upper six and lower six panels show the results of the simulations for a population inference and relationships comparison task, respectively. Each point denotes the mean accuracy for 5,000 iterations. In general, the highest group (blue) showed the best performance. The higher group (green) performed worse than the highest group, but better than the mixed group (red). However, and more importantly, we found that when only one training question was available, the highest group's accuracy in the test questions tended to decrease significantly compared to that in the training questions. In addition, the highest group's test accuracy in one training question was worse than the accuracy in 5 or 10 trainings (see Table 1; we calculated Cliff's delta using R package "effsize" [45]).

These results suggest the following: First, it is generally a good strategy to aggregate individuals with high confidence, because the highest groups tended to perform best under almost all conditions. Second, and more important, when there are few opportunities to measure individuals' confidence (i.e., highly uncertain situations such as only one training questions), an effective strategy would be to aggregate various individuals in terms of confidence levels. This is because even if individuals in a group had high confidence and performance in the training questions, the group did not always perform well in the test questions. In contrast, if a group includes both individuals with high and low confidence in one task set, their performance is less likely to decrease in another task set. This is because diverse judgments are generated by various individuals in terms of their confidence levels, which often cancels out individuals' errors. Note that in a population inference task in one training with a group size of 3 (the top left panel), the tendency of the highest group's accuracy was slightly different from that under other conditions. We speculate on this point in Discussion.

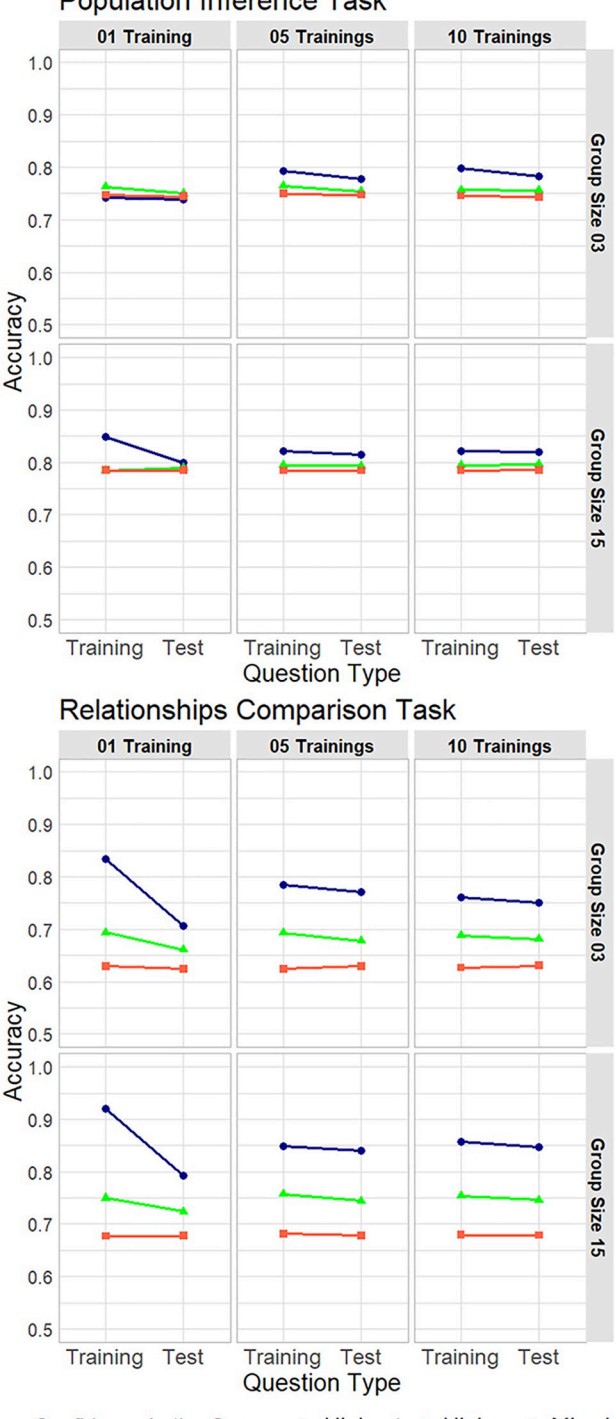

**Fig 7. Accuracy differences between training and test questions obtained from computer simulations of group judgments.** The Upper 6 and lower 6 panels show the results in a population inference and relationships comparison task, respectively. In the computer simulations, we calculated the mean accuracy of 5,000 iterations (regarded as an index of group performance). In each task, two rows correspond to two conditions of group size (3 or 15), and three columns to three conditions of the number of training questions (1, 5, or 10), respectively. The blue, green, and red lines denote the highest, higher, and mixed confidence groups, respectively.

**Table 1. Effect size (Cliff's delta) for differences of test accuracy between 1 training and 5 trainings, 5 trainings and 10 trainings, and 1 training and 10 trainings.**

| Population inference task | | 01 vs. 05 training | 05 vs. 10 training | 01 vs. 10 training |
|---|---|---|---|---|
| Cliff delta | Group size 03 | -0.235 | -0.029 | -0.266 |
| | Group size 15 | -0.120 | -0.034 | -0.155 |
| **Relationships comparison task** | | 01 vs. 05 training | 05 vs. 10 training | 01 vs. 10 training |
| Cliff delta | Group size 03 | -0.300 | -0.101 | -0.214 |
| | Group size 15 | -0.248 | -0.055 | -0.300 |

To further examine the differences between the training and test questions, we calculated "test accuracy–training accuracy" for each hypothetical group. If this value was below 0, then the group's test accuracy was worse than their training accuracy (called "test worse"). If above 0, then test accuracy was better than the training accuracy (called "test better"). As examples, we show the results for a relationships comparison task with a group size of 3 (Fig 8). (For the results for all conditions, see Supporting Information 5 in S1 File, and for the observed frequencies of training and test accuracy in the computer simulations, see Supporting Information 6 in S1 File.) Each group's difference between test and training accuracy is shown by the bars, and the proportions of test worse cases and the mean differences in test worse cases are summarized in the lower tables. We found that as members' confidence levels became more various, the number of black bars tended to decrease (i.e., in all groups, the order of values of "Test worse proportion" was "Highest > Higher > Mixed") regardless of the numbers of training questions. Furthermore, in all groups, the mean differences in test worse cases became closer to zero as the number of training questions increased (i.e., in all No. of trainings, the order of values of "Mean differences in test worse" was "1 training < 5 trainings < 10 trainings"). These results corroborate that groups comprising individuals with various levels of confidence in training questions were more likely to avoid the test worse cases, and that when

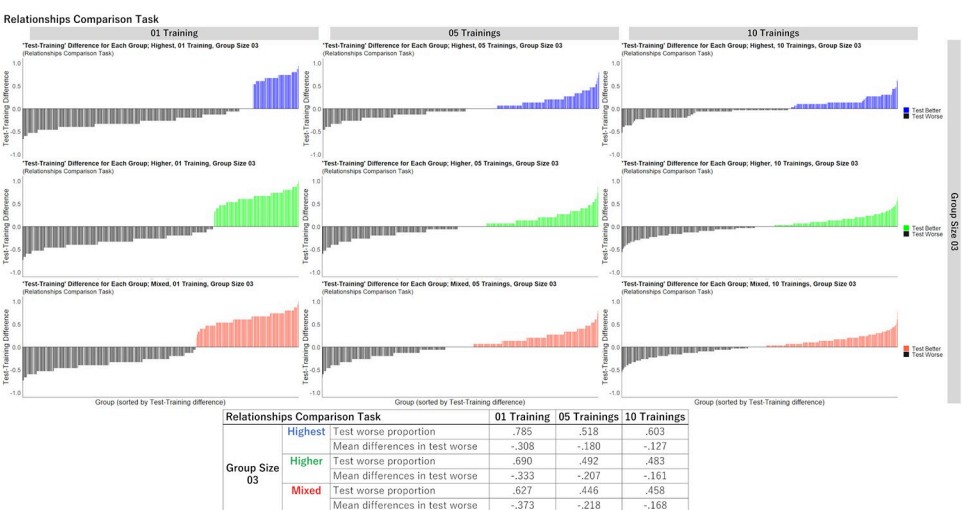

**Fig 8. Example of the difference "test accuracy–training accuracy" for each hypothetical group in the computer simulations (relationships comparison task, group size 3).** (Bar graphs) Each bar denotes each hypothetical group (5,000 groups in total; sorted by the values of the differences), and the colored bar and black bars denote "test better" (i.e., values above 0) and "test worse" (i.e., values below 0) scenarios, respectively. Upper: population inference task. Lower: relationships comparison task. (Table) Summary of "test worse proportions" (i.e., No. number of test worse cases / 5,000) and "the mean differences in test worse cases" (i.e., the mean of values denoted by the black bars).

there were only one or a few training question(s), the groups' test performance was more likely to be worse than their training performance (i.e., training performance does not predict well test performance).

Taken together, we obtained the following insights into the relationships between individuals' training confidence and groups' test performance. First, subjective confidence in one task set does not always predict performance in test tasks, because the regression coefficients of confidence in "between-questions" cases tended to be lower than those in "within-questions" cases. Of course, subjective confidence in one task set is often a good cue for identifying individuals who are good at a given task. However, it does not always predict performance in another task set. If there are few training question(s), then individuals who were "by chance" able to solve the question(s), but were in fact unfamiliar with the task domain, may be aggregated. Thus, using as many training questions as possible may be a better strategy than using only one or a few question(s). Second, particularly in highly uncertain situations, aggregating various individuals in terms of their levels of confidence is a more effective strategy than aggregating only those with high confidence. Individuals with high confidence tended to make less diverse judgments. If there are less diverse judgments in a group, then individuals' judgmental errors are unlikely to be canceled, and the group will sometimes decrease their performance in the test questions.

## 4. Discussion

In the real world, individuals must often solve "uncertain" situations (i.e., they do not know the correct answer, and it is too unclear to successfully solve the tasks) because of their lack of knowledge. In such situations, they can aggregate and rely on other people's opinions and make group judgments based on the majority rule. When aggregating others' opinions, one useful cue is individuals' subjective confidence regarding the task. In general, it seems that judgments by individuals with higher (lower) confidence will be more reliable (unreliable) and lead to group judgments in the correct (false) direction. However, even if individuals' confidence in one task set ("training questions") can well predict their performance in the same task set, this confidence cannot always predict performance in another task set ("test questions"). Rather, by aggregating individuals with higher confidence in the training questions, the performance of group judgments in the test questions may sometimes decrease drastically. This is because there is a possibility that individuals with high confidence in training questions were able to solve one or a few question(s) by chance. Our study investigated the relationships between individuals' confidence in training questions and group performance in test questions through behavioral experiments and computer simulations. Inspired by the cross-validation approach in machine learning, we developed a "training-test" approach to computer simulations. We regarded part of the questions used in behavioral experiments as "training questions" (i.e., for categorizing individuals' subjective confidence levels), and part of the remaining questions as "test questions" (i.e., to examinie the accuracy of group judgments).

The main findings are as follows. First, analyses of behavioral data revealed that individuals' confidence about their judgments in one question could well predict their accuracy in the same question ("within-questions"), but could not well predict accuracy in another question ("between-questions"). Second, the first computer simulation demonstrated that individuals with high confidence tended to make similar, less diverse judgments, and aggregating individuals in terms of different levels of confidence would likely increase the diversity of judgments. Third, the second computer simulation clarified that when there was only one training question, the accuracy of group judgments in the test questions tended to be worse than that in the training questions. In particular, the gap in accuracy between the training and test questions

was larger when aggregating individuals with the "highest" confidence (i.e., collected from the top ranks of confidence in the training question(s)) than when aggregating individuals with "higher" (i.e., collected from the above-median ranks) or "mixed" confidence (i.e., randomly collected from all ranks). These results suggest that individuals' subjective confidence in the training questions can be a useful cue for identifying individuals' abilities for tasks, but does not always predict groups' performance in test questions, especially in highly uncertain situations. Hereafter, we discuss the interpretations, implications, and limitations of our study.

### 4.1 Possible interpretations about differences in the simulation results of the two tasks

We discuss the results of the current simulation under one condition, namely training 1 with group size 3 for a population inference task. In this condition, the accuracy of the highest group in the training questions did not significantly differ from that of the test questions (top left panel in Fig 7). Although we do not have clear explanations or evidence, we speculate on the possible reasons for such differences. One possibility is the difference in the number of questions between a population inference task (70 questions) and a relationships comparison task (25 questions). This difference may have influenced our results. However, we believe that it does not, because the results observed under the other conditions were generally consistent between the two tasks (in other words, if it really influenced the results critically, more different results would be observed also under other conditions).

Another, and more important, possibility is whether or not a question directly requires exact and precise knowledge. In a relationships comparison task, participants were required specific knowledge about where the presented city is located (similar to the classification or categorization of one city into one of two countries). In such a task, subjective confidence may be more likely to reflect accuracy: if a person has knowledge about the current question (e.g., "Valencia is in Spain"), then he/she will clearly perceive whether his/her own judgment is correct and feels confident about the judgment. Thus, judgments by the "highest" groups often successfully solved the one training question even when the group size was small, but the groups sometimes could not deal well with new test questions (like "overfitting" to one training question). On the other hand, in a population inference task, participants were asked to simply compare the two presented cities and were not required precise knowledge about these cities (i.e., the exact population size). In such a task, a person will often make judgments based on equivocal or inexact knowledge about a population (e.g., not "Hitachi-shi's population, 169,816, is larger than Miyakonojo-shi's population, 158,852"; rather "Hitachi-shi may be larger than Miyakonojo-shi, because Hitachi-shi is recognizable but Miyakonojo-shi is not"; see also [37]), and the person will not clearly perceive whether his/her own judgment is correct. Thus, small groups' judgments for one training question might not perform high accuracy even in the "highest" groups, but their accuracy might not differ much between the training and test questions (like "not so overfitting" to a certain training question).

As mentioned, these interpretations are merely speculations. However, we believe that clarifying conditions where subjective confidence in the training questions can or cannot predict group performance in the test questions will improve our understandings of the wisdom of crowds in dealing with unknown uncertain tasks in the real world.

### 4.2 Practical implications

Our findings have two practical implications in terms of aggregating individuals to make more accurate group judgments.

First, when identifying individuals' confidence levels, it is better to use as many training questions as possible before dealing with the test questions. Of course, individuals with high confidence in the training questions generally have much knowledge about the tasks and often show high performance also in the test questions. However, such tendencies do not always hold true: possibly, they were able to solve the training question(s) with high confidence merely by chance, but they actually have less knowledge and are not good at the domain of the task. If many training questions for identifying individuals' confidence levels are available, then the group performance will be predicted well and the group will likely avoid such "by chance" cases in the test questions. In the machine-learning context, machines often cannot apply test data well when few training data are available because of the lack of variation in training. Our computer simulations were inspired by a machine-leaning methodology (i.e., cross-validation), and the above implication can be explained as an analogy of this problem in machine learning.

Second, if situations are highly uncertain, aggregating individuals regardless of their confidence in the training questions is an alternative strategy. In the real world, few training questions are available before solving test tasks and it is sometimes difficult to appropriately distinguish individuals' abilities. In such uncertain situations, aggregating individuals with both a high- and medium-level of confidence will be an effective strategy to avoid a decrease in group performance in the test questions. As shown in our analyses of accordance rates, individuals with high confidence tended to make similar judgments, whereas those with lower confidence tended to make more diverse ones. In other words, if many individuals with high confidence have false knowledge and make biased judgments uniformly, then the group will be led in a false direction. In contrast, if individuals have different levels of confidence, then the group will be likely to avoid heading in such a false direction because the diversity of judgments may cancel out biased judgments (i.e., wisdom of crowds). For example, in real-world group discussions, non-experts (often with low confidence) sometimes provide insightful opinions or perspectives that experts (often with high confidence) do not have. This implication may reflect such situations.

## 4.3 Limitations

This study has three limitations. The first is the effect of the task environments. In the real world, there are deceptive or "wicked" environments [32, 46] wherein individuals' confidence and accuracy are not related (i.e., individuals with high confidence sometimes make mistakes). Today, people live in a highly uncertain world. For example, three years ago, no one could predict the outbreak of the COVID-19 pandemic we are currently experiencing; and today, no one can correctly predict the number of infected people three years later. In such uncertain situations, formulating policies based on the judgments of one or few individuals with high confidence to address today's current problems may not work well for the future's unknown problems. In fact, our simulations showed that when only one training question was available (i.e., highly uncertain situation), the performance of the highest group largely decreased in the test questions (see Fig 7, especially the blue lines in "01 Training and Group Size 15"). In our study, we did not focus on the effects of task environments, and most of our experimental tasks were non-deceptive or "kind" environments wherein confidence and accuracy are related (as shown in Supporting Information 4 in S1 File). Thus, this issue needs to be examined (e.g., by setting "kind," "wicked," and "both" environments and then comparing the group's judgment performance between these environments). We believe that our arguments will be supported even in a wicked environment. This is because in a wicked environment, aggregating only individuals with high confidence will lead to irrational or inaccurate group judgments,

and mixing various individuals in terms of their levels of confidence is more likely to avoid decreasing group performance and may lead to accurate group judgments.

The second limitation is the method used to make group judgments. In our simulations, we assumed that a hypothetical group applied the majority rule. However, previous studies on the wisdom of crowds also focused on other rules to aggregate individuals' judgments [47–51]. Some studies focused on the weight of individuals' confidence ("weighted confidence" rule [52, 53]). This rule contends that members calculate the sum of the subjective confidence ratings for each alternative and then accept the alternative with the highest confidence. Consider the situation where 5 individuals respond to a binary choice question (alternatives A and B): 3 individuals chose A with confidence 20, 30, and 40, respectively, while 2 chose B with confidence 80, and 90, respectively. In this situation, if the group uses the majority rule, alternative A is chosen because the number of individuals who chose A (three) is larger than that of the individuals who chose B (two). However, if the group uses a weighted confidence rule, alternative B is chosen because the sum of the confidence for B (80 + 90 = 170) is larger than that for A (20 + 30 + 40 = 90). Previous studies also focused on the processes of making a group judgment ("transmission chain" rule [54, 55]). According to this rule, each individual's judgment is sequentially collected individually, and members update a collective solution if the individual's confidence is above a certain threshold. Consider the example above, and assume a contribution threshold as 50. First, Participant X chose A, and then the group decided on A as a collective solution. Next, if Participant Y chose B with confidence 80, the group would change the solution to B. Furthermore, if Participant Z chose A with confidence 30, the group's collective solution would remain B. Such procedures are repeated for all members, and the final collective solution is regarded as a group judgment. In the real world, it is expected that judgments with higher confidence are more likely to be accepted by group members (e.g., weighted confidence rule) or that members ask an individual's judgment one by one (e.g., transmission chain rule). As shown in the above examples, judgments made by these rules sometimes differ from those made by the majority rule. Therefore, these rules should be considered in future studies.

The third limitation is the applicability of this study to complex situations. In this study, we mainly adopted theoretical approaches using computer simulations, and therefore focused on simple judgmental situations, such as a binary choice task and the majority rule. In the real world, people will experience not only simple binary choice tasks but also more complex problems (often without a clear and objectively correct answer) and will sometimes interact with each other, such as in a discussion. Today, people can easily access and collect many individuals' opinions because of the development of information technologies. In some cases, people receive others' opinions by chatting verbally, while in other cases, people receive them only in text formats. Here, the impressions of individuals' confidence (and reliability of their opinions) in such interactions may differ from simple subjective ratings. Thus, it is necessary to investigate whether and to what extent our findings can be applied to situations outside a laboratory; for example, by using real-world social problems as experimental materials (e.g., portfolio decisions in a company), or by allowing individuals' interactions (e.g., by chat or typing).

## 5. Conclusion

Individuals often make group judgments by aggregating their judgments using the majority rule. In aggregating judgments, individuals' subjective confidence regarding tasks is a useful cue for considering whose judgment to accept. Previous studies showed that individuals' confidence can well predict their performance in one task set. In the real world, however, people often deal with not only one task set ("training" questions) but also another task set ("test" questions). We theoretically investigated the extent to which individuals' subjective confidence

in training questions could predict group performance in test questions, through computer simulations by developing a "training-test" approach (inspired by the cross-validation method). We found that confidence in a certain question did not always predict accuracy in another question (from behavioral data analyses). Individuals who had high confidence in a training question tended to make less diverse judgments between each other in test questions (from the first computer simulation). Although groups comprising individuals with high confidence in the training question(s) generally showed better accuracy, their accuracy in the test questions sometimes worsened compared to that in the training question(s), especially when only one training question was available (from the second computer simulation).

As a take home message, individuals' confidence in training questions can often become a simple and reliable predictor of a group's test performance but can sometimes become an ill predictor. In highly uncertain situations (e.g., only one training question is available), aggregating only individuals with high confidence in the training question(s) is not always an effective strategy. Rather, aggregating individuals with higher confidence and those with lower confidence in training questions will avoid decreasing group performance in the test questions. We believe that our "training-test" approach will contribute not only to expand findings on the wisdom of crowds, but will also to provide practical implications for dealing with uncertain tasks in group judgments.

## Supporting information

**S1 File.**
(DOCX)

## Author Contributions

**Conceptualization:** Masaru Shirasuna, Hidehito Honda.

**Data curation:** Masaru Shirasuna.

**Formal analysis:** Masaru Shirasuna, Hidehito Honda.

**Funding acquisition:** Hidehito Honda.

**Investigation:** Masaru Shirasuna.

**Methodology:** Masaru Shirasuna, Hidehito Honda.

**Project administration:** Masaru Shirasuna.

**Resources:** Masaru Shirasuna, Hidehito Honda.

**Software:** Masaru Shirasuna.

**Supervision:** Hidehito Honda.

**Validation:** Masaru Shirasuna, Hidehito Honda.

**Visualization:** Masaru Shirasuna.

**Writing – original draft:** Masaru Shirasuna, Hidehito Honda.

**Writing – review & editing:** Masaru Shirasuna, Hidehito Honda.

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
