## [Decision Letter · Decision Letter 0]

11 Sep 2022

PONE-D-22-20658Can individual subjective confidence in prior questions predict group performance in future questions?PLOS ONE

Dear Dr. Shirasuna,

Thank you for submitting your manuscript to PLOS ONE. After careful consideration, we feel that it has merit but does not fully meet PLOS ONE’s publication criteria as it currently stands. Therefore, we invite you to submit a revised version of the manuscript that addresses the points raised during the review process.

We look forward to receiving your revised manuscript.

Kind regards,

Chaoxiong Ye

Academic Editor

PLOS ONE

Journal Requirements:

Additional Editor Comments:

I have heard from 3 expert reviewers concerning this manuscript. Each of the reviewers comments favorably about interesting nature of the paper's primary research questions. Beyond that, as you will see, their opinions differ. I have a similar concern as Reviewer 1 about the relationship between subjective confidence and accuracy.

Reviewers' comments:

Reviewer's Responses to Questions

**Comments to the Author**

1. Is the manuscript technically sound, and do the data support the conclusions?

Reviewer #1: Partly

Reviewer #2: Yes

Reviewer #3: Partly

2. Has the statistical analysis been performed appropriately and rigorously? 

Reviewer #1: Yes

Reviewer #2: Yes

Reviewer #3: Yes

3. Have the authors made all data underlying the findings in their manuscript fully available?

Reviewer #1: Yes

Reviewer #2: Yes

Reviewer #3: Yes

4. Is the manuscript presented in an intelligible fashion and written in standard English?

Reviewer #1: Yes

Reviewer #2: Yes

Reviewer #3: Yes

5. Review Comments to the Author

Reviewer #1: The authors report the results of a study examining how different methods of aggregating people based on their subjective confidence in a training task can predict performance on future test questions. They report the results of a behavioral experiment—where subjects answered a series of binary choice general knowledge questions and made subjective confidence ratings—and then reported two computer simulations. The first simulation looked at how well-calibrated individuals were with one another. The second looked at whether subjective confidence on the training questions can predict accuracy on the test questions. The main take away from this study was that aggregating people’s judgments who are all similarly highly confident on the training questions can led to decreased performance on the test questions. The authors suggest that this occurs because sometimes people can make lucky guesses and be confident in those guesses, which doesn’t necessarily translate to good task knowledge overall (meaning they might not provide good answers later). In contrast, when there are diverse performers grouped together than a wisdom of the crowds effect is more likely to occur, meaning performance will be less likely to drop between the training and test questions.

While the basic experimental methodology seemed fine, I had two major concerns with the conclusions drawn from the data.

The first concern has to do with the limitations of the study related to materials. Throughout the manuscript there is an implicit assumption that subjective confidence and accuracy are correlated – that people who are highly confidence are likely knowledgeable. However, there are situations where confidence and accuracy are not related, such as when deceptive materials are used (Koriat & Goldsmith, 1996). In this situation people may be showing high confidence but making systematic errors – how would the inclusion of such items affect the interpretation of these results?

The second, and more important concern has to do with the interpretation and wording of the key finding: “mixing diverse individuals in a group in terms of their subjective confidence may lead to diverse judgments and achieve wisdom of the crowds.” Throughout the paper it is repeated frequently that looking at just the highly confident participants will predict lower performance on future questions, especially when there is only one test question (as shown in Figure 7). How do we know this pattern is not regression to the mean? By selecting the very confident respondents on one question, it is mathematically expected that their performance will be lower on the next question?

Relatedly, I think the conclusion of the study as worded in the abstract (and elsewhere) is misleading. Examining Figure 7, and 9 out of 12 conditions show similar performance between training and test. While three conditions do show a drop, it is only when there is one training question. So conclusions such as “we believe that our simulations following a “training-test” approach provide practical implications for keeping groups’ ability to solve future tasks and emphasize the importance of a diversity of confidence in a group” are overstated, because such diversity is only relevant when there is one training question. In all other cases picking high responders on the training set leads to accurate responses on the test set.

Finally, I thought the writing of the manuscript could be improved. There were many typos and grammatical mistakes. More importantly, there were some important issues that were glossed over:

a. In discussing the diversity of responses, it is important to clarify whether that diversity is related to the answers or the confidence in those answers (pg. 29, as one example). In particular, because the task used a forced-choice binary format, by definition there will never be much diversity in responses. People will either answer the question right or wrong. In contrast, because the confidence judgment uses a 100-pt scale it seems likely that people will display a diversity of confidence ratings. And the reasons for this confidence might vary widely. Moving forward it is important to specify whether it is the diversity of confidence or diversity of answers that is important in determining future test performance.

b. In the method section there were many choices that were not well justified or explained. For example, why did the population inference task have 70 items and the relationships comparison task only have 25? Does the difference in the number of items potentially influence the conclusions we can draw from having fewer observations?

c. Reporting effect sizes would help. On pg. 25 you write “In addition, the highest group’s test accuracy in one training question was worse than their accuracy in any other condition (i.e., training accuracy and test accuracy in 5 and 10 trainings).” When examining Figure 7, second row of panels (population inference task with Group Size 15), however, the test performance looks awfully similar across the three different training conditions. Reporting effect sizes would help readers ascertain the degree of difference.

One final note: in looking at your R analysis script there were several paths to a local file, which obviously didn’t load on my computer. As a general rule of thumb, it is good practice to ensure that others can effectively run your analysis code.

Reviewer #2: This paper investigates the group performance in answering future questions (i.e., set of relationship comparison tasks) as predicted by their (members of the same group) subjective confidence in prior questions (i.e., set of population inference tasks). For studying such a problem, the research develops a computer simulation scheme based on the empirical raw data of 200+ participants' answering those two sets of task questions. In other words, actual judgement accuracy for two types of binary question sets serve as the starting point of the studied "population", where the computer draws certain repetitive comparisons. To my understanding, insofar as me not having used such a method, it is the empirical form and meaning of those original questions, as well as the way they were answered, that delimits what kind of conclusions and significance can be drawn.

While the statistical method/process itself seem intact and rigorous, and the initial motivation/idea of the research problem very interesting, I would argue that a few things may need to be clarified or explored further:

1. The idea of diversity may need more clear explanation. From line 63, the author explains "Especially in inferences of binary choice situations, when the mean of individual accuracy is above 0.5 and individuals make “diverse” judgments among them (i.e., various patterns of errors in questions)...". In a way, a simpler explanation may be, "diversity means not everyone makes the same mistake (different people make different mistakes)".

2. The meaning of "predicting performance of question 'i' with the confidence of answering question 'j'". It may be hard for the reader to begin to recognize the significance of such prediction. In other words, why would we ever want to predict performance of a relationship comparison task with the confidence when answering an (unrelated) inference task? On the other hand, it is not a new or unintuitive idea that there is some power when predicting the accuracy of the current question with the confidence for the same binary fact-based question. More descriptions need to go to portraying the relationship between such two types of questions as selected by the research, which should also accompany more reflection on the generalizability of the results. For example, the study often uses the distinction of "prior questions" and "future questions", but the results may be only applicable to predicting the performance of one specific type of task with another. The term "prior" and "future" may be too grand.

3. Boundary conditions of the study design may be explored and discussed further. As suggested at the beginning, it is the empirical raw data in the first place that delimits the viability and applicability of the computer simulation results. One should recognize the inherent features of the original questions to better understand the boundary conditions of the results and the conceptual model. For example, what may happen if the tested questions (e.g., if we intentionally sample such questions) happen to elicit diversified answers but the wrong answer by majority count? In another sense, the choice of the task type etc. may already dictates the end result of the computer simulation, but too general conclusions are made with no condition specification. Eventually, sometimes group wisdom works and others it may not.

Reviewer #3: The authors proposed a simulation study based on a human behavioral dataset of different individuals answering to trivia questions and reporting their confidence levels. The simulation study is formulated into a forward prediction problem where a majority voting model is given the data of a pool of individuals, their historical answers and their confidence levels and asked to take those confidence levels into account for a future task of question answering. They observe that it improves the prediction over not taking the historical confidences into account. And the more diverse the sampled individuals in the dataset, the better the performance.

The question is well motivated and the structure of the paper is easy to follow. The hypothetical examples are very convincing. The results are also quite interesting.

The referee, however, held reservations towards the main statement of the paper. After all, this is not a human prediction task that the authors are evaluating against. The main argument in the paper was stated, "These results suggest that if people aggregate diverse individuals in terms of subjective confidence in prior questions, they will likely avoid decreasing performance in future questions." However, the word "people" here is problematic. This is because, unlike the study in reference [33], in this work they didn't have any human subjects, in an experimental setting, to be given a pool of past decisions and confidence levels from other individuals, and then be asked to make the decision based on them. Instead, what was investigated appears to be simply how a majority voting model can improve its performance in a forward prediction task if taken into account a diverse pool of historical samples rather than a homogeneous one.

This sentence at line 130 doesn't fully make sense and requires clarifications: "Because some individuals with high confidence may be overconfident or could solve prior question(s) by chance, their errors are likely to be canceled out by mixing individuals with lower confidence into a group (individuals with low confidence may have lack of knowledge and therefore sometimes make random guesses. Thus, their judgments may differ from judgments by individuals with high confidence, which leads to diverse judgments in the group)." It appears that, not matter how confident an individual is, facing a lack of knowledge, they will make random guesses one way or another (either due to over confidence, or due to lack of confidence). If that is the case, wouldn't having a group with either a homogenous or a diverse confidence levels landing the same level of judgement diversity?

And again, the wording "prior" and "future" can be problematic. On top of the concern with this not about how to predict the decision making behaviors of human individuals (but actually of a much simplified majority voting model), this is not a forward prediction task that the authors are evaluating against, since there is no notion of "time" or "history" here. Spliting a dataset into training and test sets doesn't imply that one has a temporal precedence over the other.

In summary of the past three points, the referee suggests the authors to modify their main arguments and clarify these points, and potentially rephrase their titles and abstracts to correct for controversial terms (e.g. "prior", "future").

Other minor comments:

Fig 6. Please provide legend for different colors in the figure.

6. PLOS authors have the option to publish the peer review history of their article (what does this mean?). If published, this will include your full peer review and any attached files.

Reviewer #1: No

Reviewer #2: No

Reviewer #3: **Yes: **Baihan Lin

---

## [Author Response · Author response to Decision Letter 0]

11 Oct 2022

We appreciate your careful review of the previously submitted manuscript. We found the comments reasonable and helpful in improving our manuscript. We send Responses to Editor and three Reviews.

In the revised manuscript, yellow highlighting indicates the sections with major revisions.

---

## [Decision Letter · Decision Letter 1]

14 Nov 2022

PONE-D-22-20658R1Can individual subjective confidence in "training" questions predict group performance in "test" questions?PLOS ONE

Dear Dr. Shirasuna,

Thank you for submitting your manuscript to PLOS ONE. After careful consideration, we feel that it has merit but does not fully meet PLOS ONE’s publication criteria as it currently stands. Therefore, we invite you to submit a revised version of the manuscript that addresses the points raised during the review process.

The manuscript has been evaluated by two reviewers, and their comments are available below.

One of the reviewers has raised a minor concern regarding their suggestion of additional detailed text that relates the research to real or hypothetical social phenomena. 

Could you please carefully revise the manuscript to address all comments raised?

We look forward to receiving your revised manuscript.

Kind regards,

Alice Coles-Aldridge

Editorial Office

PLOS ONE

 Journal Requirements:

Reviewers' comments:

Reviewer's Responses to Questions

**Comments to the Author**

1. If the authors have adequately addressed your comments raised in a previous round of review and you feel that this manuscript is now acceptable for publication, you may indicate that here to bypass the “Comments to the Author” section, enter your conflict of interest statement in the “Confidential to Editor” section, and submit your "Accept" recommendation.

Reviewer #2: All comments have been addressed

Reviewer #3: All comments have been addressed

2. Is the manuscript technically sound, and do the data support the conclusions?

Reviewer #2: Yes

Reviewer #3: Yes

3. Has the statistical analysis been performed appropriately and rigorously? 

Reviewer #2: Yes

Reviewer #3: Yes

4. Have the authors made all data underlying the findings in their manuscript fully available?

Reviewer #2: Yes

Reviewer #3: No

5. Is the manuscript presented in an intelligible fashion and written in standard English?

Reviewer #2: Yes

Reviewer #3: Yes

6. Review Comments to the Author

Reviewer #2: While the previous comments have been addressed well, I would personally like to see a bit more text that relates the research to certain real or hypothetical social phenomena with some more details filled. It can be either in the conclusion part or the limitation section. With this said, if other reviewers are fully satisfied with the revised manuscript, I hold no objection as the study is technically sound and argumentatively clear.

Reviewer #3: The referee would like to thank the authors for the revision and detailed response to the reviews. The revision and response address most of my concerns.

7. PLOS authors have the option to publish the peer review history of their article (what does this mean?). If published, this will include your full peer review and any attached files.

Reviewer #2: No

Reviewer #3: **Yes: **Baihan Lin

---

## [Author Response · Author response to Decision Letter 1]

24 Nov 2022

We appreciate your careful review of the previously submitted manuscript. We found the comments reasonable and helpful in improving our manuscript. In the revised manuscript, yellow highlighting indicates the sections with major revisions (e.g., where we have added a new discussion or explanation).

For our responses, please see the attached file ("Response to Editor/Reviewer").

---

## [Editor Report · Decision Letter 2]

12 Jan 2023

Can individual subjective confidence in "training" questions predict group performance in "test" questions?

PONE-D-22-20658R2

Dear Dr. Shirasuna,

We’re pleased to inform you that your manuscript has been judged scientifically suitable for publication and will be formally accepted for publication once it meets all outstanding technical requirements.

Kind regards,

Yann Benetreau

Staff Editor

PLOS ONE
---

## [Editor Report · Acceptance letter]

27 Feb 2023

PONE-D-22-20658R2 

Can individual subjective confidence in training questions predict group performance in test questions? 

Dear Dr. Shirasuna:

I'm pleased to inform you that your manuscript has been deemed suitable for publication in PLOS ONE. Congratulations! Your manuscript is now with our production department. 

Kind regards, 

on behalf of

Dr. Yann Benetreau 

Staff Editor

PLOS ONE